# Defining a Digital System for the Pedestrian Network as a Conceptual Implementation Framework

**Mona Jabbari** [1,2,*] **, Zahra Ahmadi** [1] **and Rui Ramos** [3]

1   Department of Business Administration, Faculty of Education and Business Studies, University of Gävle,
    801 76 Gävle, Sweden; zahra.ahmadi@hig.se
2   CitUpia AB, 104 30 Stockholm, Sweden
3   Urban and Regional Planning, Department of Civil Engineering, University of Minho,
    4710-057 Braga, Portugal; rui.ramos@civil.uminho.pt
*   Correspondence: mona.jabbari@civil.uminho.pt

**Abstract:** In cities today, the digitalization of mobility is one of the most crucial tools that link each single mobility service providers (e.g., ride sharing, public transportation, air travel, etc.) to users. Based on the smart pedestrian network model, the purpose of this study is to initially provide the requirements towards the digitalization of a pedestrian network model and subsequently to draft an institutional framework towards the effective implementation and management of pedestrian mobility that will develop/create a pedestrian network as a new structure in the city. The methodology is applied in three phases, with three separate approaches: "desk approach" for a data gathering standard that is knowledge-based and connected to walkability; a "digitalization approach" for citizen and stakeholder participation in policy co-creation; and a "business approach". A business approach is defined as a set of operations that takes one or more types of input and produces a customer-valued outcome. In this case, customers are citizens and the business approach by applying a digital system is assessing policies and finding/defining an optimized combination of shared applicable/effective policies to implement the pedestrian network. By boosting an innovative linkage of these three phases, digitalization of the pedestrian network has great potential to improve the walkability planning process and therefore to create more sustainable and livable urban spaces.

**Keywords:** pedestrian network; digital system; urban planning process; citizen; business strategy

## 1. Introduction

Since the turn of the millennium, human activities have led to a transformation of landscapes and territories, changes in ecology and their interconnections, as well as enormous scientific and technological progress [1,2]. The presence of components and natural values in the city today is a necessary requirement for the urban territory's environmental rehabilitation [3]. The walking and urban systems are part of the same space whose integrated management is a condition of the sustainability of territories and cities. Moreover, walkability within the urban fabric represents an increase in environmental quality, which urban planning should reinforce and improve [2,4,5].

Stakeholders as urban authorities provide more significant leverage for handling digital system planning processes than national authorities, and they often have better expertise and awareness of the local environment and the implementation of efficient digital policy for the area [6–10]. Urban digital system planning is not a distinct field, but rather an umbrella term that covers everything from technical substitution in digital urban planning to urban space. Nevertheless, digital planning and urban planning are usually the responsibility of different sections in the municipality; they therefore have their own planning processes. Individual planning procedures may come up with distinct and often opposing policies, resulting in a less effective influence on cities in terms of achieving climate targets and decreasing coordination.

In cities today, digitalization of mobility is one of the most crucial tools that links each single mobility service provider (e.g., ride sharing, public transportation, air travel, etc.) to users [11–13]. Digitalization helps city planners with new business models and the implementation of various strategies towards the development of socio-economic dynamics and long-term sustainability in smart cities. Nonetheless, there is still no appropriate digitalization for pedestrian mobility, and policy makers have not yet provided an institutional framework for pedestrian mobility. Such digitalization is a necessity not only for more sustainable cities but also for pedestrian safety and security. The present study aims to identify a digital urban planning process related to walkability in the city that transfers to other cities and is capable enough to improve: (a) accounting for walkability measures appropriate to the specific city culture, structure, and their likely development; (b) considering and linking all significant stakeholders in the planning process; and (c) integrating behavioral, technical, operational, organizational, and financial issues. The methodology's backbone is made up of three separate but complementary phases, which is novel since it is a well-thought-out mix of well-established procedures in a single integrated methodological framework. Below, the study explains more:

Phase 1: The "desk approach" provides a rough logistic city profile. This task is performed using info on the city and its walkability characteristics. Phase 2: The "digitalization approach" refines, improves, and transforms the policies selected by using a digital system. It collaborates with citizens and stakeholders and engages them in co-creating policies. It is a governance model method for incorporating them into a sustainable urban mobility plan (SUMP) framework, resulting in the definition of a common policy component thanks to active/fruitful participation of important stakeholders in a long-term/integrated planning process. Phase 3: the "business approach" emphases the most suitable behavioral stimuli capable of favoring the policy implementation/adoption, based on distinguished yet integrated state-of-the-art policy valuation methodologies (e.g., behavioral and business model analysis) coupled with the digital system. Hence, it provides policymakers with an efficient, effective, and innovative decision-support system on a B2C (business-to-consumer) basis.

Medium-sized and large cities are already allocating more space for walking in the road network, especially after the COVID-19 crisis [14–16]. Moreover, knowing the characteristics of inhabitants relevant in the walk mode is necessary for improving the conditions, creating an initial incentive for the population to change their behavior. The suggested methodology contributes to the identification and improvement of effective pedestrian network solutions in smart cities and implements a smart pedestrian network project. Bringing together knowledge achievement, policy co-creation, and behavior change analysis within a single methodological approach, the methodology aims to identify an optimal policy package. Applying a digital system provides better conditions for the pedestrian as an end-user, and urban digital system development has a lot of potential to achieve climate goals. Furthermore, this system supports close contribution of urban plans in the urban department and gives grounds that create an integrative approach in the municipal digital planning process taking urban plans into account.

The remainder of this paper is arranged as follows: After reviewing the smart pedestrian network concept in Section 2, we explain case studies and define the methodology of the paper in Sections 3 and 4. Finally, in Sections 5 and 6, we present the discussion and conclusion.

## 2. Integrative Approaches for Creating a Smart Pedestrian Network

In the 1990s, the term "smart city" was used to describe the use of information and communication technologies (ICT) and modern infrastructure in cities [17–19]. The construction of smart cities has become a popular urban development plan. The central idea behind making a "smart city" is to provide improvements to all modes of transportation that might transform it into a better place for transit riders as well as pedestrians and thereby cover one of the smart city's main goals. Walkability is a primary focus among the smart city elements that create transformative change in towns and cities as well as

making it possible to achieve the three goals of the smart city. In addition to a multitude of health benefits due to walking, there are many economic benefits for developers, employers, and retailers [20,21]. After all, walking has the lowest carbon emissions, does not pollute the environment, is the cheapest and most reliable mode of transportation, and is a great social leveler.

In recent decades, important planning approaches have considered the planning process as their main focus of theorization, research, and practice. Urban planning attempts to improve urban living conditions by making efficient use of land while maintaining a balance between urbanism and nature [1]. Such concerns present a larger impact in promoting walkability in the city. In part, walkability is known as a type of "green transportation" which has a low level of environmental influence, conserving energy without any air and noise pollution [22]. In recent years, the travel behavior community has paid a lot of attention to understanding social networks and social interactions. Finally forming bonds and engaging in social engagement provide ties to the overall well-being and happiness of individuals in the society [23]. Van Cauwenberg, De Donder [24] found that the frequency of walking could have an impact on the number of social interactions between neighbors, as walking increases the likelihood of both spontaneous and planned interactions. In addition, Weijs-Perrée, van den Berg [25] showed that people who walk more regularly are more happy with their social life. Walking frequency was found to have a direct and indirect positive effect on the number of social interactions and social pleasure. Similar to walking, cycling perhaps leads to more natural social interactions than using a car as a transport mode. Furthermore, people who have more social contacts may walk and cycle more frequently [26].

Achieving these social goals is just one dimension of urban planning that can be reached by walkability in the city. Making a whole city more walkable will also support enormous economic growth. Economists and policy wonks rarely mention walking as a strategy for economic development [27]. Individually, there is a range of studies around cost-benefit and the economic benefit of dropping heart disease through walking and through physical activity [28]. Moreover, there is another cost-benefit in public space by creating central business districts (CBDs) in the city [29]. In CBDs in large cities like Sydney and Melbourne, walking is the predominant way to get around [30].

The pedestrian network is a new structure in the urban area that consists of cohesive streets and provides suitable features for walking. The personal, social, economic and environmental benefits of walking are well-documented: walking reduces traffic congestion and pollution; it is beneficial to individuals' health and well-being; it provides economic benefits; it has an impact on real estate prices and enhances the sociability and vitality of urban spaces [31–35]. For these reasons, the concept of pedestrian network has been placed at the center of various urban policies and has been a main topic for urban designers and planners over the last decades. The smart pedestrian network (SPN) project [36] highlights the relevance of the three dimensions of urban planning, IT, and city marketing with reference to theory, practice, and applications.

Worldwide, the types of planning processes differ and depend on the country's legal system, planning systems, and institutional frameworks as well as the economic, social, and cultural norms of individual places. They are important in order to guide urban progress in the majority of cities in the world [37,38]. Digitalization as a new approach in the process of urban planning has been an important topic of discussion in recent years and includes different types of electronic innovation improvements [39] that collect vast amounts of data, provide connectivity, and analyze "big" data [40]. Technology development entails benefits in terms of supporting increased customer intimacy, increased cooperation within markets, and growth in open innovation [41]. Organizations need to adopt these new technologies in order to stay competitive [42]. However, related studies show that data collection and technology are only the raw material [43]. It is not enough for firms to implement digital technologies; focus should be on reconfiguring the whole organization to take advantage of the information these technologies enable [41,44]. Thus, the technology can be

integrated across people, processes, and functions to boost competitive advantage [45]. All development in the organization is dependent on economic conditions in the society [27], for instance whether digitization is included in such development strategies. In fact, urban planning has a significant role to play in assisting governments and civil society to face the urban challenges of the 21st century.

Current technology-driven implementations in urban planning are an essential step; it is nonetheless critical to remember that the major actors are people and the human dimension of cities. The concept of a participatory innovation system, in which citizens and communities interact with public authorities and knowledge creators, is critical in this regard. User-centered innovation governance models are co-designed as a result of this collaborative interaction. The urban transformation addressed transformational leadership in which individuals are the primary "drivers of change" through their empowerment and motivation that assures key municipal concerns, such as sustainable behavior and green development [9,10,46–48]. It is vital to adapt to the new social, technological, and spatial context in which we live in order to blend the key actors of cities with spatial forms and processes. Moreover, the integrative approaches concept was influenced by a survey applied in two cases studied (Porto and Bologna).

## 3. Concept of Study

This lack of success in technologically novel suggestions can be attributed to the user's lack of motivation as well as to a lack of knowledge with tools. According to this context, the goal of this research is to fill the gap by looking at how driven B2C (business-to-consumer) e-consumers are to apply apps for micro-mobility. To do this, a survey was conducted that involved 573 people in Porto and 865 people in Bologna under the support of the "Smart Pedestrian Network" project and the co-funding Smart Urban Futures (ENSUF). The project aims to develop a pedestrian navigation tool to assist pedestrians in choosing routes based on specific criteria. The project team carried out a questionnaire for collecting data about people's perceptions of the conditions provided by the city and their behaviors. The purpose of the survey and the range of questions were previously explained in individual working sessions. The questionnaire, which was completed in 2019, used a semi-closed question style and included six main segments with a total of 43 questions. The survey was carried out in the two cities (Porto and Bologna) based on diverse parts, including: pedestrian profile; frequency and purpose of walking; walkability criteria evaluation; assessment of the overall walking conditions provided by this street and neighborhood; pedestrian behavior and preferences; and mobile applications for walking. In this paper we highlighted the need/demand for citizens to apply digital applications in their own activity (walking).

Figure 1 shows different groups' percentages in their replies to the survey in terms of sex, age group, and type of resident in both cities. The sixth part of the survey was related to the mobile applications for walking. The following survey question was discussed: Would you consider using any mobility application in the future? Results from users in both cities that showed around 65% of users did not plan to use mobility apps in the future. This feedback may show that respondents would like to use apps that are more accurate, practical, and familiar for pedestrian navigation [49]. Furthermore, this shows another knowledge gap, namely that there is insufficient e-consumer awareness, need/demand, and motivation for a sustainable system.

How urban and digital planning processes should be cohesive is an ongoing research field and several authors have worked on this subject with diverse approaches and serious reviews [44,46,50–52]. The previous integrative techniques all have one thing in common: they all use digital system models. Digital system models are frequently used to aid decision-making in digital planning in order to fulfill the user's goal, such as minimizing total system cost, total emissions, and so on, within specific sectors and time periods. A large number of prior research has used digital system models to explore effects in several areas under various situations at the urban scale [8,50,52].

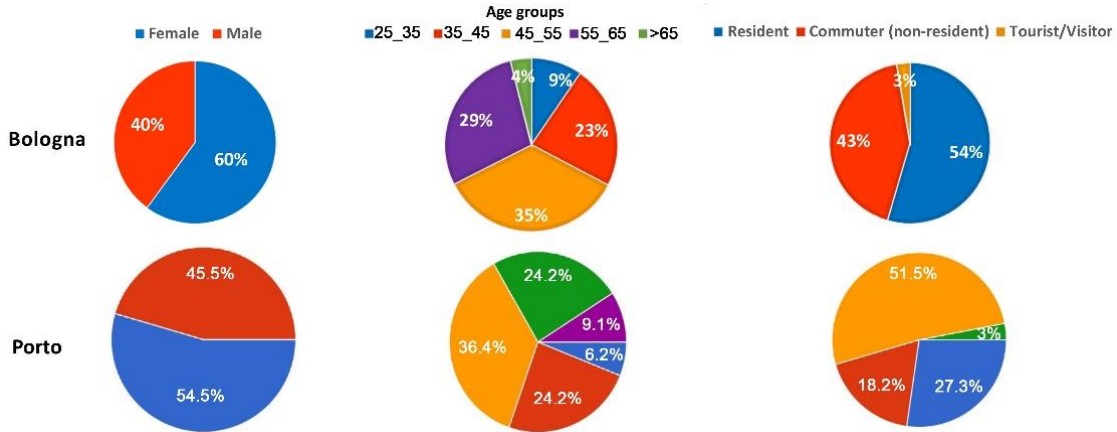

**Figure 1.** Charts included different groups of respondents in the SPN project survey.

## 4. Methodology

This paper's methodology uses three approaches as the urban transformation in urban mobility, including the "desk approach" for a data gathering standard that is knowledge-based and connected to walkability; the "digitalization approach" for citizen and stakeholder participation in policy co-creation; and the "business approach", as well as how they are integrated to implement a smart pedestrian network as shown in Figure 2. This framework has a tendency to emphasize social/technological/economic factors. Previous research and EU-level declarations and efforts have highlighted the necessity for an integrated approach to understand both digital and urban planning for an effective mobility transition. The requirement recognition impacts the urban and digital departments' practical planning processes, and how they should be merged is not expressly articulated.

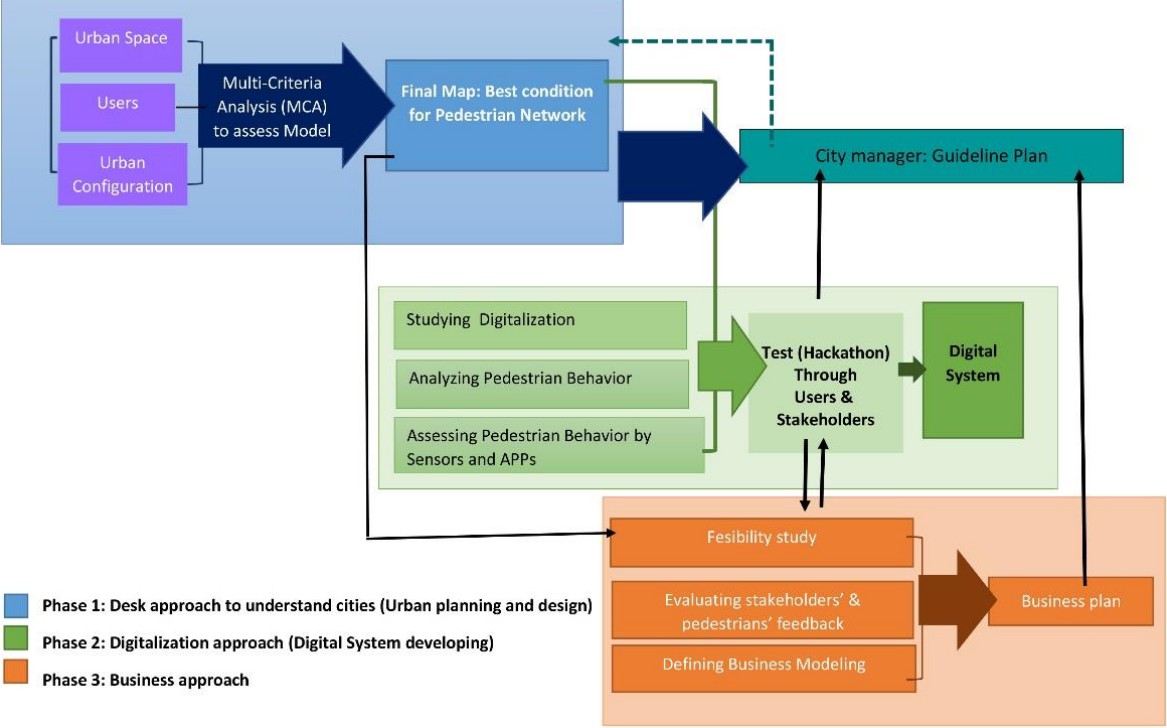

**Figure 2.** Methodology framework for the digitalization of pedestrian network.

This paper emphasizes the importance of combining interconnected solutions that complement one another in order to address urban and digital planning concerns in a complete and dynamic manner. It presents a systematic framework for describing the many

and different aspects of city planning. The goal of this research is to provide a methodology for incorporating certain characteristics of the urban planning process into a spatially explicit digital system model for municipal mobility planning, with both digital and urban planners participating. Whereas prior studies tend to reflect integrating digital aspects into the urban planning process, this paper considers the converse, i.e., integrating features of urban planning into a digital system model for a pedestrian network. The abovementioned displays an extension of urban factors influencing the urban and digital planning process for pedestrian networks.

### 4.1. Desk Approach to Understand Cities

The urban planning process to create a pedestrian network should be focused on infrastructure development. The process assesses and identifies an optimum pedestrian network that considers citizens' opinions and demands (especially for children, women, and differently abled people), as well as how to access and connect pedestrians to a pedestrian network. To achieve the optimum pedestrian network, the built environment, urban configuration, urban functions, accessibility, and natural environment parameters are analyzed [34]. These parameters have been evaluated differently in terms of the scale of analysis (micro to macro) and context city. It is based on survey research to understand better pedestrians' opinions and demands in terms of physical and social aspects. In this phase, GIS has been an increasingly adopted tool to make spatial analysis, while space syntax has also been used to evaluate the configuration and street connectivity [4,34,35,52–55].

### 4.2. Digitalization Approach

In order to influence people's behavior and service usage, smart cities must develop a dynamic, demand-based pricing model which leads to prices for Mobility as a Service offerings to be considered high [12]. The operation of smart technology can be beneficial if it can quickly detect or monitor various spatial or unusual events happening in and around the city. ICT, telecom, and advanced network infrastructure deliver better e-services to their stakeholders [10,56]. The sensors and cameras provide computer resources for managing, storing, and exchanging a high quantity of Internet of Things (IoT) data, as well as back-end capabilities for data analytics and control, through the internet and remote processing [57].

Several digitalization approaches to the construction of the pedestrian network dataset have been proposed. These range from the manual digitalization of sidewalks or other pedestrian facilities over aerial or satellite imagery where sidewalks and footpaths are discernible, to the application of geoprocessing analysis using available data as a basis to automatically generate the pedestrian network. Automated methods include buffering the street centerlines [58] or using the geometry of city blocks [59,60]. The study applies a comprehensive digitalization approach to foster citizen and stakeholder engagement in implementing a smart pedestrian network by creating a digital system in the municipality. The digital system has a strong and rapidly growing evidence base that can be used to promote walkability in the city. This is an effective system to create confidence to implement the pedestrian network by citizens in the future that is managed by the municipality as the main stockholder in the city [61].

Moreover, an innovative approach influences the urban components by connecting to apps and the digital system, and it creates new place-making for a pedestrian network. By recognizing the streets that are more and less walkable and the particular causes of this, the system can be supportive to make a communication platform as a "hackathon" between users and stakeholders to improve walkability and develop streets of the pedestrian network in the city. It is evident that the digitalization approach encourages people's habits to improve the walkability in the city. Finally, it makes better mobility habits of citizens and underlines the active role of citizens in this process [2,8,44,62].

*4.3. Business Approach*

Business is a significant behavioral activity and is related to ecological improvement. In B2C (business-to-consumer), transportation accounts for the largest share of greenhouse gas emissions [63]. As the amount of e-consumers rises, so does interest in sustainability studies that are related to these kinds of transactions, as sustainability remains one of humanity's toughest challenges [64]. The number of consumers who have purchased goods via e-commerce in 2020 was approximately 3.47 billion people, equivalent to about 44.5% of the global population. During the COVID-19 pandemic in 2020, the total value of the global B2C e-commerce market was about USD 2.44 trillion [64,65]. People's shopping behaviors are linked to their social communities via e-commerce. Much work has been done using social network data to promote product and service sales. However, in order to improve the quality of city life, there is less knowledge about the impact of digital activities on social interactions. There is a large service and activity around the pedestrian network as a significant business hub that joins citizens, stockholders, and the city to enhance the livable space. A feasibility study filling gaps between the digitalization design and walking operation in the pedestrian network is presented in this phase. The feasibility study determines how the intervention of a digital system serves to implement the pedestrian network in a smart city dynamically. Such assessment influences and promotes walking behavior, improves well-being for pedestrians, and increases performance in city management. Finally, this phase provides policymakers with a well-organized and effective decision-support system. The second point is stakeholders' and pedestrians' feedback related to applying the digital system, which is important in the phase. The digital system is tested by applying simulation in real space to identify values in a prototype business model that shows more potential to develop it. Their assessment can contain quantitative or qualitative analysis collection. All feedback will be considered in the digital system as key points for further improvement.

## 5. Discussion and Final Remarks

In Figure 2, we created a digitalization of our pedestrian network framework in relation to urban transformation in order to show how stakeholder agendas and intentions correspond to citizen/user demands/needs. As a result, technological advancements must be supported with methodological advancements, which, above all, increase organizational and user motivation. In this way, it serves as a delivery method for businesses to get closer to their end users. New collaborative design processes are required, which are customized to the new social, technological, and spatial context in which we live. Hence, the core aspects related to the implementation of this conceptual model are as follows.

*5.1. Integrating Policies and Citizens' Demands in Strategic Urban Planning via Collaborative Participation*

Presently, the ways we communicate have been changing and adapting to new digital systems that frequently include characteristics such as mobility, interaction, and interconnection. Considering citizens' demands related to the pedestrian network obtained by analyzing semantic, temporal, and spatial patterns leads to improvement of the process of urban design and management. On another hand, municipality influences citizens' motivation to achieve smart mobility goals through communication, and creates role models. Furthermore, it produces more motivation for citizens to develop further the pedestrian network as the main actors. Municipalities should be able to incorporate policies with citizens' demands to develop more sustainable mobility projects with users, especially those with disabilities and different generation mindsets. The need to incorporate projects in a digital system used by the public is essential. It will allow for the possibilities presented by these new technologies to be described as "creating a new sort of reality, one in which physical and digital settings, media, and interactions are woven together throughout our daily lives".

### 5.2. Increasing Knowledge and Understanding of Pedestrian Network as New Structure

Relationships describe the architecture of how to create a structure, including potentials, system design, know-how, and partnership data and knowledge. The pedestrian network will be built, operated, and maintained by citizens and municipalities in smart cities. The digital system of the pedestrian network is implemented by the municipality as a local team and could rely on a single local technician to ensure that the pedestrian network is maintained properly and minor repairs are undertaken. For example, part of a highly connected street of the pedestrian network is damaged and made unsafe for pedestrians, causing a pause in pedestrian movement. The local technician was trained in how to repair and restart the pedestrian street movement in such a case. The pedestrian journey meets with the bigger challenge presented in terms of street safety, security, or even movement barriers as well as interfaces with public transportation. Whenever the pedestrian network needs serious technical repairs, such requirements will inform the local team by the app, and then the municipality will fix it.

By understanding the user experience, we also become aware of the connection process and communication system related to managing changes and developing the pedestrian network based on citizens' opinions. Furthermore, the digital system interfaces and engages with more citizens across different user groups (disability, women, and elderly). There is a mode of citizen interaction through feedback within the pedestrian network that will impact the business plan. Finally, all of these factors define a guideline of the pedestrian network in order to create better-quality urban space in the city as a new structure.

### 5.3. Digitalization of a Smart Pedestrian Network and Assessment in Terms of the Urban Environment

Business innovation, digitalization, and sustainable development issues have become increasingly significant to researchers. However, the lack of literature combining business innovation with design thinking in the urban planning process, on the other hand, shows that studies have been gradually deviating from the backbone of what is design thinking. For example, the uncertainties surrounding the COVID-19 epidemic, the deterioration of the ecological environment, the impending global economic impact, and widespread social instability all present complex and difficult situations for urban development. This study highlights that the urban planning process must be combined with digitalization and business plan as a design thinking framework. Design thinking for the urban planning process should be combined with digital technologies to analyze big data and improve business model innovation. The use of data mining algorithms is supportive for design thinking to describe pedestrian characteristics more correctly, to find and understand citizen needs more deeply, and to make up for the deficiency that puts too much emphasis on the human being. Finally, by providing open data sources it is possible to improve the interaction between citizens and systems, and the municipality will create a pedestrian network as a new structure city based on citizen opinion that is managed by the municipality.

One of the aids for integrating the urban and digital planning process at the municipal level is that it supports a coherent urban digital system planning process by addressing the issue of local mobility transition. This approach can be achieved by integrating features of the urban planning process into the digital system model. In particular, this paper showed where the involvement of municipal stakeholders can be integrated into the process. Further, the opinion users in Porto and Bologna show that integrating spatial plans and open data features can lead to the identification of new pedestrian networks and mobility demands to apply different types of micro-mobility in each district. This study developed an innovative methodology of integrating the urban and digital planning process into mobility system models, and the study can be applied to other municipalities for urban mobility system planning by being customized to their own local contexts.

## 6. Conclusions

Presenting the three approaches in this topical collection has the added value of giving the readers a wide understanding of the complex set of issues related to the digitalization of a pedestrian network (smart pedestrian network). While also giving specific survey results from case studies (Proto and Bologna), it mentioned the performance of apps related to walk mode, for citizens or e-customers. Developing a smart pedestrian network requires as an important element to have a sustainable transport system and specifically an integrated digital system with special attention to citizens.

Even if our experiences thus far have been positive, this study is simply the first step toward understanding what benefits are possible by applying a digital system for the smart pedestrian network in the context of the B2C e-customer. In future research, it will be necessary to study more cases in a more systematic manner, for example, by following an urban development project from conception to completion. Interviews with stakeholders and citizens could be a useful addition to the data collection strategies we employed in this study. Another interesting study would be to combine stakeholders and citizens with the digitalization in a B2C e-customer context to see how this could influence stakeholder and citizen recognition and contribution from the start of the project as a dynamic approach. In addition, we suggest that local management (municipality) are suitable locales in which to create a digital system in order to develop the pedestrian network that follows from urban transformation.

**Author Contributions:** Conceptualization, M.J.; methodology, M.J.; software, M.J.; validation, M.J., Z.A. and R.R.; formal analysis, M.J. and Z.A.; investigation, M.J., Z.A., and R.R.; resources, M.J., R.R.; data curation, M.J.; writing—original draft preparation, M.J.; writing—review and editing, M.J. and Z.A.; visualization, M.J. and Z.A.; supervision, Z.A.; project administration, M.J.; funding acquisition, R.R. All authors have read and agreed to the published version of the manuscript.

**Funding:** This research received no funding.

**Institutional Review Board Statement:** Not applicable.

**Informed Consent Statement:** Not applicable.

**Data Availability Statement:** Not applicable.

**Acknowledgments:** The study partially summarizes the results of a PhD thesis entitled "Combining Multi-Criteria and Space Syntax Analysis to Assess a Pedestrian Network: An Application for the Pedestrian Network of Porto (Portugal) and Qazvin (Iran)", supervised by Rui Ramos, the Center for Territory, Environment, and Construction of the School of Engineering of University of Minho. ''SPN: Smart Pedestrian Net" Project is special funding provided by JPI Urban Europe titled Smart Urban Futures; and co-financing by FCT- Fundação para a Ciência e Tecnologia (ENSUF/0004/2016) between 2017–2020.

**Conflicts of Interest:** The authors declare no conflict of interest.

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
