# Peer review of "Defining a Digital System for the Pedestrian Network as a Conceptual Implementation Framework"

_sustainability, doi:10.3390/su14052528_

Round 1

Reviewer 1 Report

Dear authors,

your paper deals with a very actual topic, smart cities and walkability, which is for sure of interest. Nevertheless, I have many important remarks. I will write them in the following and hope that they will help you improve your future works.

  1. First and foremost: it seems that you would like to present a new approach to create smart pedestrian networks, but this new approach is not explained throughout the paper.
  2. No practical application of this approach is given. Have you tried to implement it in any city? Are there some parameters/variables that allow to judge the improvements of such a method?
  3. In general, the paper looks like more as a review of existing approaches towards the overmentioned topic. That is fine, but you should then propose it as review paper, and write it as such. In this case, from the very beginning the reader waits to see and understand your new approach, but without success.
  4. Also going through the single sub-sections related to the three steps of the method, the reader does not have the feeling of understanding a new approach: it seems to be more a list of what has been done, the advantages of these models/techniques, and the benefits of their application. The same is for the discussion and conclusions. Once more: this is totally fine, if you decide to write a review paper. But I don’t think this is the case…
  5. This comment is related to comment 2: in the last section “Future research” you mention that you will analyse “more cases in a systematic manner”, but in this manuscript I cannot see one example of these concrete cases, and this is a major concern for me.

Minor comments:

  1. Line 11: “available pedestrian network mode” – which one? Here it would be good to be a bit more specific.
  2. Line 57: capacity – probably you meant capable.
  3. Lines 176-185: which are the previous 5 parts?
  4. Line 183: the following survey one question…
  5. Figure 1: could you provide a higher quality figure? Percentages are difficult to be read.
  6. Line 240: pricing model? Maybe the concept should be better explained.
  7. Line 298: a framework of the the digitalization…

Author Response

Your paper deals with a very actual topic, smart cities and walkability, which is for sure of interest. Nevertheless, I have many important remarks. I will write them in the following and hope that they will help you improve your future works.

  1. First and foremost: it seems that you would like to present a new approach to create smart pedestrian networks, but this new approach is not explained throughout the paper.

Smart Pedestrian Network (SPN) defined in the paper text. In fact, the paper detected gaps to implement the smart pedestrian network project and present solutions to improve process of the smart pedestrian network.

  1. No practical application of this approach is given. Have you tried to implement it in any city? Are there some parameters/variables that allow to judge the improvements of such a method?

We carried out one survey in the two cities (Porto and Bologna) based on diverse substances, including  Pedestrian profile; Frequency and purpose of walking; Walkability criteria evaluation, Assessment of the overall walking conditions provided by this street and neighborhood; Pedestrian behavior and preferences; and Mobile applications for walking. In this paper we highlighted need/demand citizen to apply digital application in the own activity (walking).

  1. In general, the paper looks like more as a review of existing approaches towards the over mentioned topic. That is fine, but you should then propose it as review paper, and write it as such. In this case, from the very beginning the reader waits to see and understand your new approach, but without success.

The authors didn’t agree with the suggestion made by the reviewer.  Although Approach is not innovative you can find it in the energy and public transportation topics. But it applied in the smart pedestrian network project is unique and innovative that there are a lot of gaps. The authors clarified them by this paper.

  1. Also going through the single sub-sections related to the three steps of the method, the reader does not have the feeling of understanding a new approach: it seems to be more a list of what has been done, the advantages of these models/techniques, and the benefits of their application. The same is for the discussion and conclusions. Once more: this is totally fine, if you decide to write a review paper. But I don’t think this is the case…

The paper was rewritten following the structure and the topics suggested by the reviewer. The authors improved the link between the topics addressed and clarified several points to support the methodological decisions adopted.

  1. This comment is related to comment 2: in the last section “Future research” you mention that you will analyse “more cases in a systematic manner”, but in this manuscript I cannot see one example of these concrete cases, and this is a major concern for me.

Have done

Minor comments:

  1. Line 11: “available pedestrian network mode” – which one? Here it would be good to be a bit more specific.

This part was also rewritten and Clarified, Please check it

  1. Line 57: capacity – probably you meant capable.

Have done

  1. Lines 176-185: which are the previous 5 parts?

Clarified it

  1. Line 183: the following survey one question…
  1. Have done
  1. Figure 1: could you provide a higher quality figure? Percentages are difficult to be read.

Have done

  1. Line 240: pricing model? Maybe the concept should be better explained.

Clarified it

  1. Line 298: a framework of the the digitalization…

Modified it

Reviewer 2 Report

Dear editor. I read the paper and recomend some changes. 

  1. Novelty of the work should be mentioned in the abstract.
  2. Introduction should be improved by more related references.
  3. Methodology must be mentioned in a section seperated.
  4. Quality of the figures should be improved.
  5. Conclusion section should be in the paper.
  6. Some new references can be mentioned to enhance the paper belong 2021.

Author Response

I read the paper and recommend some changes. 

The sections following the structure and the topics suggested by the reviewer.

  1. Novelty of the work should be mentioned in the abstract.

Mentioned it in the abstract. Please consider it

  1. Introduction should be improved by more related references.

Have done

  1. Methodology must be mentioned in a section separated.

Have done

  1. Quality of the figures should be improved.

Have done

  1. Conclusion section should be in the paper.

Have done

  1. Some new references can be mentioned to enhance the paper belong 2021.

Updated more

Round 2

Reviewer 1 Report

Dear authors,

I am still not totally convinced of the soundness of the manuscript. You roughly tackled some of my comments, but without providing me with any explanation.  Nevertheless, you partially improved the paper.

My last suggestion is to let your case study emerge more, reserving to it a whole section. In this way, it will be easier for the reader to understand how your methodology can be effectively applied. 

Best regards

Author Response

Dear Reviewer

Thank you for your comment.

Would you please check Author's Response Letter and the manuscript in the attachment?

Best regards

Mona Jabbari

Round 3

Reviewer 1 Report

Dear authors,

thank you for keeping your time and provide much clearer and better structured responses. 

I do not have any more suggestions and wish you all the best for your present and further publications.

Best regards